# OpenReview forum: "Unified Preference Optimization: Language Model Alignment Beyond the Preference Frontier"
_TMLR — Accepted by TMLR_

### Review · Reviewer_qKkf · 2025-02-11

**Summary Of Contributions:**

This paper introduces a novel technique for LLM alignment by introducing offline advantage estimation using value functions. The experimental results show the advantage of the proposed method.

**Audience:**

Yes

**Claims And Evidence:**

Yes

**Requested Changes:**

1. Add discussions and comparisons to related works including [1, 2, 3].
2. I would like to see experiments on alignment in additional dimensions beyond readability and safety, such as verbosity and helpfulness as considered in [2, 3] on standard alignment benchmarks such as AlpacaEval. This would help better compare the proposed method and baselines.
3. (Optional) The models considered are kind of outdated. Experiments on more SOTA models (e.g., Llama 3) will support the claim better.

[1] Dong et al. "SteerLM: Attribute Conditioned SFT as an (User-Steerable) Alternative to RLHF."\
[2] Want et al. "Arithmetic Control of LLMs for Diverse User Preferences: Directional Preference Alignment with Multi-Objective Rewards."\
[3] Zhang et al. "Reward-Augmented Data Enhances Direct Preference Alignment of LLMs."

**Strengths And Weaknesses:**

Strengths:
1. The paper is clearly written and well-motivated.
2. The experiments show that the proposed method can achieve optimal tradeoff between readability and safety.
3. The analysis also supports the method well.

Weaknesses:
1. The experiments only consider preferences in two dimensions: readability and safety. It would better support the method if the authors could experiment in more general settings beyond these two dimensions.
2. The results presented are restricted since there is no standard alignment leaderboard performance that is reported, which raises concerns about whether the method, especially considering the additionally learned value function, can contribute a lot compared to simpler methods such as SteerLM [1], Directional Preference Alignment [2], and Reward-Augmented Alignment [3].

[1] Dong et al. "SteerLM: Attribute Conditioned SFT as an (User-Steerable) Alternative to RLHF."\
[2] Want et al. "Arithmetic Control of LLMs for Diverse User Preferences: Directional Preference Alignment with Multi-Objective Rewards."\
[3] Zhang et al. "Reward-Augmented Data Enhances Direct Preference Alignment of LLMs."

---

> ### Author Response · Authors · 2025-02-16
> **Response to Reviewer qKkf (part 1)**
>
> Thank you for the constructive feedback, and we apologize for the delayed response. We perform additional experiments and have made corresponding changes in the main text and appendix of the paper.
>
> >The experiments only consider preferences in two dimensions: readability and safety. It would better support the method if the authors could experiment in more general settings beyond these two dimensions.
>
> >such as verbosity and helpfulness
>
> Based on your suggestion, we have added a “verbosity” objective to Appendix A.4.6. With this objective added as well, we show even larger improvements compared to before. For computational reasons, we have only performed this evaluation on LLAMA-13B for DRO-V, since it generally performs better than aoPPO in Figure 6 and presents the best multi-objective performance on our GPT-4 evaluation (Table 2).
>
> However, we would like to mention that we already include 3 primary types of objectives (excluding verbosity) and 8 total objectives: preference, readability, and multiple safety categories. These are fairly unique objectives that do not overlap. The preference objective largely targets helpfulness, not the others (e.g., OASST specifies that “factual accuracy and helpfulness are first and foremost” in their prompt for classifying an agent’s reply; Kofp et al., 2023). As shown in Appendix A.4.8, the chosen/“helpful” response can be quite unsafe.
>
> Within safety, we maximize each of the 6 different safety rewards (e.g., identity attacks, sexually explicit, etc.) through optimizing of the “worst-case” safety category (i.e., maximin optimization). We opt for this aggregation function because the presence of **any** safety violations presents an issue, so a sum of such rewards is inappropriate. That said, this still optimizes all 6 of the reward functions (simply from a worst case point of view). In Figure 5 and 6, we demonstrate that we show improvement across all of these categories as well.
>
> After your proposed change, we are optimizing 9 total “rewards” (preference, 6 forms of safety, readability, and verbosity), showing improvements across all of these. Please let us know if you have any suggestions for other objectives to further showcase UPOs capabilities.
>
> > especially considering the additionally learned value function can contribute a lot compared to simpler methods such as SteerLM [1], Directional Preference Alignment [2], and Reward-Augmented Alignment [3].
>
> >Add discussions and comparisons to related works including [1, 2, 3].
>
> Per your suggestions, we have added discussions to the related work and in Appendix A3.3.3. We find that these techniques largely fall into existing categories of techniques we already touch upon. Further, we do not believe that these methods are necessarily simpler. Our value function takes less than 0.5% of the compute time, adds less than 5 lines of code (given it’s a 2-3 layer MLP), and simply estimates an expectile of the reward (e.g., for $\tau=0.5$, it just computes the mean reward).
>
> We briefly discuss the other techniques below; as far as we can tell, **all of these are on-policy techniques (or at least only demonstrate as such in their experiments), which we do not evaluate due to their poor computational efficiency (at least 2-3x slower to train vs. offline methods) and because offline techniques scale better for the same computational cost (Appendix A.3.3)**. They also all fall into the issues outlined in Section 4.1.
>
> We note that SteerLM learns an “attribute prediction model”, similar to a value or reward model, and requires a lot of steps: learning the attribute prediction model, perform attribute-conditioned SFT, bootstrapping to select “high quality samples” and on-policy sampling, followed by another round of SFT. As for Directional Preference Alignment (DPA), it performs on-policy sampling and rejection sampling, followed by behavioural cloning (BC) or SFT (step 3 on page 6). We previously noted the development of rejection sampling with Khaki et al. (2024) and Liu et al. (2023). Beyond the inefficient on-policy generation, **both SteerLM and DPA are forms of k% BC, which is commonly used as a baseline in offline RL literature, and unfortunately, falls within the same issues as detailed in Section 4.1 with ULMA and other methods**. What constitutes “good enough”? What if none of the recommendations are perfect/”good enough”, but many are “okay”? Our technique is able to handle that efficiently by moderately weighting moderately good generations, highly weighting great generations, etc. This is one of the core hypotheses of why we believe MODPO does not perform well on our evaluation.
>
> Reward-Augmented Alignment combines goal-conditioning, binary reward margins (as already discussed in related work), and DPO. However, again, any binary techniques mean that it will result in the exact fundamental issue outlined in Section 4.1. Also, it largely focuses on on-policy data, which we don’t consider for evaluation (Appendix A.3.3).

---

> ### Author Response · Authors · 2025-02-16
> **Response to Reviewer qKkf (part 2)**
>
> We continue our responses below.
>
> >The results presented are restricted since there is no standard alignment leaderboard performance that is reported
>
> >beyond readability and safety, on standard alignment benchmarks such as AlpacaEval.
>
> Based on your suggestions, we have employed the AlpacaEval (2.0) prompt and evaluation framework to evaluate the quality of our generations on our own evaluation data. For similar cost reasons, we only employ the best multi-objective baseline (DRO-V) for this evaluation. We observe similar relative results as our original GPT-4 evaluation in Table 2 in Table 12 (Appendix A4.4.7). If you would like us to run any other leaderboard evaluations or with other datasets beyond this, please let us know.
>
> **However, aggregated across techniques, we are able to detect a significant length bias in AlpacaEval based on these evaluations, which we show in Figure 10 (Appendix A4.4.7), corroborated by Dubois et al., 2024 and others**. With a statistically significant correlation of 0.9 between average length relative to chosen and win % (p < $10^{-11}$) on LLAMA-13B, we believe the bias is quite notable. Hence, we believe that such evaluation frameworks may not be an unbiased representation of the holistic and overall quality that one would expect from a chatbot.
>
> For more context, our justification for why we originally excluded any leaderboards is provided in Appendix A.4.5. We believe the best type of evaluation in LLMs can at most be relative, with as many controlled elements in the environment, training, and evaluation, as possible. We believe we have done this to our best ability within computational limits. Absolute evaluations of any technique are challenging given the stochasticity (at scale) at every step of the way (parameter initialization, sampling from training data, noise in numerical precision in bfloat16, etc.).
>
> Further, each leaderboard has its own (unknown) biases that may be completely different from the biases that we wish to reinforce in the auxiliary objectives. Just as we discovered, AlpacaEval has been discovered to have a length bias favouring longer outputs (Dubois et al., 2024). In fact, Wang et al., (2024) show that even AlpacaEval-2.0 has a pretty noticeable length bias (Figure 5). **Would AlpacaEval prefer a long, unsafe generation over a shorter yet safe generation?** It is unclear exactly how that is controlled, and given that, we initially preferred a simpler and more unbiased solution (relative to what we train on).
>
> Additionally - **we would like to mention that we did perform some true human evaluation of our prompt and found results indicating that GPT-4 performance was correlated with the true human evaluation given our prompt** (human rankings were quite similar, though they preferred aligned methods more than GPT-4). We did not present this result in the paper as it introduced additional complexity to the submission process, but if it improves your confidence in our GPT-4 metrics in Table 2, we can certainly include it.
>
> We believe we have performed thorough evaluations, with: 5 models, 2 model types, 3 datasets, 9 baselines (4 multi-objective, 5 single objective), and 8/9 objectives, alongside breakdowns of model performance by safety category (Figure 6b-6f), safety reward (Figure 6a), and safety proportion (Figure 4). In the appendix, we examine readability (Appendix A.4.2) and safety (Appendix A.4.5) in more detail and tradeoffs in objectives (Appendix A.4.4). We have added evaluations on verbosity (Appendix A.4.6) and AlpacaEval (Appendix A.4.7).
>
> Dubois, Yann, et al. "Length-controlled alpacaeval: A simple way to debias automatic evaluators." arXiv preprint arXiv:2404.04475
>
> >(Optional) The models considered are kind of outdated. Experiments on more SOTA models (e.g., Llama 3) will support the claim better.
>
> We address this in Section 5.1 and Appendix A.3.3. One of the primary concerns with newer models is that they employ many different techniques to “pre-tune” or “pre-align” the models for safety, as demonstrated in these sections. Specifically, LLAMA-3 and other new models like Gemma from Google incorporate significantly greater sets of safety measures, with red teaming, dataset sanitization, and other techniques to ensure safety. Given that safety is one of our alignment tasks, we believe that this is similar to “task leakage” in the training step for these LLMs and makes post-hoc safety alignment more redundant. Further, we believe that these models themselves have (a) not grown significantly larger since 2023 (LLAMA-3 is presented at similar model scales) and (b) not changed much given the transformer architecture is largely similar to before. Hence, at the computational cost of ~2,000-4,000 GPU hours, we did not perform this additional experiment with LLAMA-3.
>
> Thank you for reading, and please let us know if you have any other feedback or concerns.

---

> ### Author Response · Authors · 2025-02-27
> **Follow-Up on Responses**
>
> We hope you are having a great week! Thank you again for your constructive feedback on our paper, and we believe that these suggestions have improved the quality of our work. To summarize, we have added:
>
> - an additional "verbosity" objective, alongside preference, safety categories (with 6 separate rewards), and readability; across all of these **9 different rewards, we show notable gains over prior multi-objective approaches with Pareto fronts**
> - additional evaluation with the evaluation framework of AlpacaEval 2.0, which shows (a) relatively similar results to our own GPT-4 evaluation and (b) a significant length bias ($r \approx 0.9, p < 10^{-11}$), corroborated by many past work, and we can show human evaluation (which we originally omitted for simpler submission) to validate our GPT prompt
> - comparison to related work you suggested, all of which use (computationally expensive) on-policy sampling, which we discussed previously and have added to in Appendix A3.3.3 and our experimental setup; further, some of these techniques require multiple stages, rejection sampling, and even more (we show in Appendix A.3.3 that for the same compute cost, offline techniques are still better or as good)
>
> If you have any additional thoughts on our responses (i.e., Response to Reviewer qKkf (part 1/2)), please do let us know. We are happy to discuss and incorporate any additional feedback, and we look forward to hearing from you.

---

> > ### Comment · Reviewer_qKkf · 2025-03-06
> >
> > Thank the authors for the clarification. The responses addressed most of my concerns. However, I don't think the argument "all of these are on-policy techniques (or at least only demonstrate as such in their experiments), which we do not evaluate" is accurate enough. In my opinion, [1] and [3] are both proposed primarily for offline scenarios, e.g., given an SFT or preference dataset that contains (fine-grained) scores as labels, their methods are very similar to conditional SFT or DPO. Can the authors explain why they did not compare them experimentally, e.g., the setup differences?

---

> > > ### Author Response · Authors · 2025-03-06
> > > **Response to Reviewer**
> > >
> > > Thank you for the response. Below is our reply to your question about past work, with more revisions to Appendix A.3.3.
> > >
> > > - We simply cannot train on-policy approaches without significant horizontal scaling (e.g., **SteerLM uses 16x more GPUs and up to 32x more GPU memory compared to any models we train/evaluate**) and massive costs (e.g., **renting 16 p4de nodes like SteerLM would result in costs of $15K per day**).
> > > - If trained offline, **Zhang et al. (2025) is similar to MODPO, which we perform experiments with (and CDPO, which we cite)**.
> > > - To our knowledge, offline techniques sample prompts **and generations** from datasets, whereas on-policy sampling demands generating from the LM, e.g., SteerLM requires on-policy, autoregressive generation from the LMs (like RLHF), even if the prompt is sampled from an offline dataset. This makes them at least 3x more expensive (as discussed in Appendix A.3.3).
> > > - **Nearly all multi-objective techniques (including MODPO, UPO, DRO-V) are based on weighted, conditioned, or filtered versions of SFT/DPO.** From a certain perspective, UPO applies a weighted SFT objective, where the weight is simply the baselined reward. However, practical properties like stability, efficiency, and performance are important. Unlike Dong et al., (2024), UPO doesn't require multiple steps beyond SFT and alignment, sampling from LMs, or 128 A100 GPUs. UPO needs no more forward passes through the LM, whereas every on-policy technique requires many generation steps (prefill, decode).
> > >
> > > Below is a deeper dive:
> > >
> > > **Dong et al., (2024) [1]**
> > > 1. **There are a significant number of steps for learning and they must be performed sequentially (not end-to-end), taking up significant compute time.** This involves learning a separate "attribute prediction model" (which is effectively a custom reward model), performing SFT conditioned on these attributes (which differs from standard SFT; hence, it cannot share these SFT models with other techniques, reducing computation), followed by on-policy top-$k$ sampling and another round of SFT. These are significantly more steps than MODPO, UPO, KTO, etc., and requires significantly more compute.
> > >
> > > 2. **In Step 4 of SteerLM, they state usage of on-policy sampling from the LM (similar to RLHF).** Even though the prompts are sampled from the offline dataset, their responses are sampled **on-policy** (from the LM policy itself). As detailed in Appendix A.3.3, even sampling 1 response for a "reasonable" hardware setup can result in weeks of training.
> > >
> > >     The authors of this work mention that they use 16 nodes for training these models (16x more than our "reasonable" compute power), which is what makes this scale of on-policy sampling possible despite being orders of magnitude more expensive than DPO. Below are quotes from their work indicating the use of on-policy sampling.
> > >
> > > >**By sampling the policy network**, RLHF effectively navigates the response space of language models and identifies responses that are of various qualities. The response samples are subsequently utilized to influence and shape the behavior of language models according to their reward values. In Step 4 of STEERLM, the objective is to accomplish a similar objective by leveraging the Attribute Conditioned SFT and Attribute Prediction models from the previous steps.
> > >
> > > >By combining v′ with prompts from the same training datasets, **we use the top-k (=50) sampling to generate multiple responses**
> > >
> > > **Zhang et al. (2025) [3]**
> > >
> > > 1. **Offline reward-augmented DPO with binary margins (as in Zhang et al., (2025)) is very similar to MODPO, which we already include in evaluations and discussions.** CDPO (which we cite) also employs goal-conditioning similar to this approach. Moreover, we discuss the "binary reward margin" approach in detail in the related work, Section 4.1, and the appendix. Ultimately, these techniques are flawed as we point out in Section 4.1: what constitutes "good enough" to make a response $y_w$ versus $y_l$? Learning in a strict binary regime can collapse in these cases.
> > >
> > > 2. **Their technique has no clear and thorough offline evaluations for distinct objectives (like readability), whereas MODPO (similar approach) does.** Since reward-augmented DPO is the tagline for MODPO as well, so we opted to compare to MODPO, which is (a) more established for offline multi-objective alignment, (b) simpler to understand, (c) easier to code up, and (d) has pre-existing multi-objective evaluations. We've added this reasoning to Appendix A.3.3.
> > >
> > > 3. **Though the technique differs from MODPO when it's on-policy (i.e., as presented in much of Section 6.1), we cannot work with/reproduce this given our computational constraints and lack of efficiency in generation.** We also believe comparisons between offline and on-policy techniques are unfair given the vast difference in compute between the two approaches, as discussed in detail in Appendix A.3.3. For the same cost, we could vastly scale up offline methods.

---

> > > > ### Author Response · Authors · 2025-03-25
> > > > **Follow-up to Response**
> > > >
> > > > We hope you are having a great week! Thank you again for your continued feedback on our paper, and we believe that these have served as a large contribution in the improved quality of our work. To summarize our previous response to your concerns:
> > > > - Regarding "on-policy" methods, we class them as any method requiring generation from the LM policy (i.e., generating/sampling under the training policy), whereas offline methods don't sample from the LM under training. **SteerLM is on-policy, requiring LM generation, 16x the GPU compute, 32x the GPU memory, and over 20x the monetary cost of training.** Generally, as we mention in Appendix A.3.3, on-policy training requires at least 3x more compute and cost (even more if for rejection/bootstrap sampling) and it can take weeks with the reasonable hardware limitations that we have.
> > > > - **Offline, Zhang et al. (2025) is similar to MODPO (based on DPO and binary reward margins), which we already perform experiments with and improve upon.** Zhang et al. (2025) doesn't have thorough offline multi-objective evaluation on distinct objectives; hence, we prefer MODPO since it is more established as a multi-objective technique. The distinction to MODPO is its utility in on-policy training, for which they perform several experiments. As previously mentioned, we do not evaluate that scenario.
> > > > - **Nearly all multi-objective techniques (including MODPO, UPO, DRO-V, others) are based on weighted, conditioned, or filtered versions of SFT/DPO.** From a certain perspective, UPO applies a weighted SFT objective, where the weight is simply the baselined reward. However, practical properties like stability, efficiency, and performance are important. Unlike Dong et al., (2024), UPO doesn't require multiple steps beyond SFT and alignment, sampling from LMs, or 128 A100 GPUs. UPO needs no more forward passes through the LM, whereas every on-policy technique requires many generation steps (prefill, decode).
> > > >
> > > > Please let us know if you have any additional feedback or concerns regarding our work. We are happy to discuss and incorporate any additional feedback, and we look forward to hearing from you.

---

> > > > > ### Comment · Reviewer_qKkf · 2025-03-25
> > > > >
> > > > > Thank the authors for the detailed response. My concern regarding the comparison with SteerLM is addressed, as the proposed method is more offline compared to the online SteerLM, which needs more computation. However, my concern regarding the lack of a simple offline baseline that is similar to Zhang et al. (2025) still remains. What if we just train the LLM by conditioning on the reward labels using the vanilla DPO? In my understanding, this is the conditional DPO counterpart to the CSFT baseline that you compared and is different from MODPO, which changes the objective.

---

> > > > > > ### Author Response · Authors · 2025-03-26
> > > > > > **Response to Reviewer qKkf**
> > > > > >
> > > > > > Thank you for the continued feedback. Regarding this baseline, that is a great suggestion and a combination of conditional learning (like CSFT) with the DPO objective can definitely enable multi-objective learning in theory. We have added a section to the appendix, Appendix A.4.10, explaining the procedure and algorithm with a quantitative comparison of evaluation metrics and further discussion.
> > > > > >
> > > > > > Specifically, we have implemented and experimented with this using a minor adaptation of DPO with a "reward token" added to the prompt during training, specifying the reward of the associated generation (rounded to the nearest hundredth, i.e., 0.00, 0.01, 0.02, ..., 0.99, 1.00) in text like CSFT. Beyond this augmentation to the prompt, there are no additions or changes to DPO, and we call this method Conditional DPO (ConDPO). For computational cost, we did not run this for all model sizes and only did LLAMA-13B, which is the largest and strongest model.
> > > > > >
> > > > > > We believe the results are fairly conclusive: ConDPO seems to improve upon the GPT-4 evaluation score of DPO and seems to be similar to MODPO, but it shows nearly identical metrics to DPO and MODPO in terms of multi-objective optimization (safety and readability). The improvement margins for UPO (with either DPO or KTO as a base method) and DPO, MODPO, or ConDPO in terms of these auxiliary objectives is just as big, showing that our proposed multi-objective technique, UPO, outperforms all of them.

---

> > > > > > > ### Comment · Reviewer_qKkf · 2025-03-26
> > > > > > >
> > > > > > > Thank the authors for the response. This addresses my concerns.

---

### Review · Reviewer_Ceaz · 2025-03-13

**Summary Of Contributions:**

This paper proposes a novel preference alignment method for large language models, termed Unified Preference Optimization (UPO). In contrast to prior work, it addresses the complexity and scalability issues of existing alignment methods. For instance, the widely used RLHF suffers from inefficiency and high training costs, while DPO, although more efficient, struggles with aligning multiple objectives effectively. The proposed UPO method leverages a weighted MLE optimization objective (implemented with KTO as the foundation, as detailed in the Appendix), unifying both preference objectives and auxiliary objectives. This alignment approach is not only concise but also offers advantages in terms of training cost, reward mechanism design, and data requirements. In the experiments, the authors use Reading Level and Safety as auxiliary objectives to validate the feasibility and alignment effectiveness of UPO. The experimental results demonstrate that UPO, when evaluated by GPT-4, matches or surpasses baseline alignment methods in terms of general generation quality, while also showing superiority in the two auxiliary objectives compared to multi-objective alignment methods. Additionally, the analysis experiments support the low training cost and efficiency of UPO.

**Audience:**

Yes

**Broader Impact Concerns:**

The authors did not include a Broader Impact Statement. A potential risk of UPO is that in cases of extreme adversarial use, such as deliberately inducing harmful generation during training, how can we balance model alignment with social responsibility? (This might also be a derivative issue of Weakness 4.)

**Claims And Evidence:**

Yes

**Requested Changes:**

1. In response to Weakness 1, please add an explanation for $D_{KL}$ about Equation 1.
2. In response to Weakness 2, please revise Figure 2 to ensure consistent capitalization, preferably using uppercase for the first letter.
3. In response to Weakness 3, to better demonstrate this advantage, I suggest improving and modifying Table 5 to include multi-objective baselines and other resource consumption metrics.
4. In response to Weakness 4, could there be a "collapse point" in complex real-world applications?

**Strengths And Weaknesses:**

**Strengths**
1. From a methodological design perspective, UPO effectively balances the flexibility of traditional RL with the simplicity of DPO, addressing the challenge (i.e., conflicting auxiliary objectives) of optimizing auxiliary objectives in multi-objective alignment. As shown in the analysis section, for the auxiliary alignment tasks discussed in the paper, it requires only a short training time and simple value head architecture (see implementation details in A.3.4).
2. The practical application of UPO is straightforward and easy to implement, requiring no additional paired data or complex auxiliary parameterized models, making it particularly suitable for resource-constrained scenarios.
3. The experimental design in the paper is thorough and comprehensive, including error bars for multiple experiments and providing significance tests. The analysis experiments also discuss hyperparameters, which helps validate the generalization ability. The model architecture choices cover multiple variants from two model families (Llama and Pythia) under 20B parameters. Although these may not be the most recent open-source models, the justification provided by the authors is reasonable.
4. The authors provide both theoretical derivation and empirical results, particularly in the appendix, where some open discussions may be of interest to the community. The paper is well-written, and the case studies offer intuitive insights into the motivation behind the work.

**Weaknesses**
1. The symbol $D_{KL}$ in Equation 1 is not explained in the context. Although researchers in the community might intuitively guess its meaning, it would be better to include an explanation. I confirmed that it refers to the standard KL regularization term only after referring to the cited work [1].
2. In Figure 2, the capitalization of the first letter in the font is inconsistent. For example, "**P**rompt" is capitalized, while others are not (like "**p**reference **d**ataset").
3. In Section 5.4, the authors mention the efficiency of training but lack a comparison with other multi-objective preference optimization baselines. I understand that this result is advantageous, but empirical evidence is necessary. Additionally, I would like to know if there are advantages in VRAM consumption during forward and backward propagation.
4. On page 9, the authors state, *"We believe this is an important use case since not all designer preferences or dispreferences may be directly reflected in collected datasets"*. I strongly agree with this point, but it is worth noting that when scaling to more diverse auxiliary objectives (as in the added section A4.6), how can we ensure that the general generation quality is not compromised?


Reference:

[1] Mohammad Gheshlaghi Azar, Mark Rowland, Bilal Piot, Daniel Guo, Daniele Calandriello, Michal Valko, and Rémi Munos. A general theoretical paradigm to understand learning from human preferences.

---

> ### Author Response · Authors · 2025-03-15
> **Response to Reviewer Ceaz**
>
> Thank you for your in-depth review of our paper. We have responded to your concerns and feedback below, with revisions in our paper in Figure 2, Section 3, and Appendix A.4.8/A.4.9.
>
> >The symbol $D_{KL}$ in Equation 1 is not explained in the context. Although researchers in the community might intuitively guess its meaning, it would be better to include an explanation.
>
> Thank you for pointing this out, and we can confirm that we are referring to the standard KL divergence/regularization term. We have added a revision to Section 3 which details the mathematical contents of Equation 1 in words ("the preference reward subject to a KL-divergence penalty $D_\mathrm{KL}$") with an added reference to the original paper from Kullback & Leibner (1951).
>
> >In Figure 2, the capitalization of the first letter in the font is inconsistent. For example, "Prompt" is capitalized, while others are not (like "preference dataset").
>
> We have fixed this inconsistency and every label in Figure 2 is now "Title Cased".
>
> >In Section 5.4, the authors mention the efficiency of training but lack a comparison with other multi-objective preference optimization baselines. I understand that this result is advantageous, but empirical evidence is necessary. Additionally, I would like to know if there are advantages in VRAM consumption during forward and backward propagation.
>
> This is a great point. Since we couldn't add the results for all methods in Table 5 due to lack of space (additionally because we are at the 12 page limit), we have added an extended computational analysis in Appendix A.4.8 in Table 13. While it's impossible to "save" any time or memory compared to DPO (which doesn't optimize for auxiliary objectives) or MODPO (which uses no additional parameterization), we show that our solution is the most consistently resource-efficient of offline RL-based multi-objective solutions. For simplicity, we evaluate PYTHIA-6.9B and LLAMA-13B, which are the largest models of each LM family, and PYTHIA-1.4B, the smallest model. We observe that DRO-V uses the most average memory with 25-50% more time spent in the RL component relative to UPO and aoPPO uses the least memory yet with several orders of magnitude higher time spent due to an additional LLM forward pass.
>
> >it is worth noting that when scaling to more diverse auxiliary objectives..., how can we ensure that the general generation quality is not compromised? ... could there be a "collapse point" in complex real-world applications?
>
> As far as our experiments go, we have not encountered such a collapse point wherein the generation quality is compromised; our GPT-4 evaluation score has always remained within $\pm$ 4-5 of the values shown in Table 2 (and the minimum for UPO on LLAMA-13B is around 42.0 with maximums around 49-50). However, this is always within or slightly above the confidence interval margin of $\pm$ 4.4, so it is reasonable to conclude that even a "soup" of 9 unique reward functions (across which we improve all of them, as shown in Appendix A.4.6 and Section 5) with a preference objective does not seem to detrimentally affect generation capabilities.
>
> However, it is possible that there is an unknown "collapse point" given too many objectives (perhaps 50+, 100+, ...), rendering the overall reward too noisy, or perhaps those which interact with each other in some sort of adversarial manner. In this case, any multi-objective system would find it difficult to optimize it. While we are unable to come up with 50 or 100 objectives, given the interesting premise, we took a computationally easy but potentially challenging use case of optimizing for a "random" set of weighted rewards (i.e., where $R(x, y)$ is uniformly random between 0 and 1).
>
> While we optimize for a random reward in Appendix A.4.9, we see that generation quality is unaffected (even slightly better). While that may seem counterintuitive, we provide an explanation in the text that shows that (a) a random reward leads to a noisy SFT objective with an average (advantage) weight coefficient around one and (b) both the preference objective and the implicit KL-divergence penalty always keep us from having poor generation quality.
>
> **Consequently, let's consider the absolute worst case for UPO: the coefficients for the (random) RL objective $\gamma \to \infty$ and hence, the preference objective is not even used.** Even in this case, we default to (noisy, weighted) SFT-like behaviour with reasonable, non-random responses. This is because we always retain a weighted log likelihood objective on the target responses, and even if the weights are random, we will **always** upweight the log likelihood of the chosen and rejected responses by a non-negative value (i.e., $\gamma \exp(c)$ for some $c \in \mathbb{R}$).
>
> >The authors did not include a Broader Impact Statement.
>
> Thank you for pointing this out. We've included this on Page 13 detailing some of the risks we perceive with misuse of UPO and generally, alignment.

---

> ### Author Response · Authors · 2025-03-23
> **Follow-up to Response**
>
> We hope that you had an excellent weekend! Thank you again for your constructive feedback on our paper, and we believe that these suggestions have improved the quality of our work. To summarize:
>
> - Regarding Weakness 1 and 2, we have added an explanation for $D_\mathrm{KL}$ for Equation 1 and accordingly revised Figure 2 as required.
> - Regarding Weakness 3, we have added a thorough analysis of training time usage and GPU memory usage for aoPPO (as implemented by Ethayarajh et al., 2024) and DRO-V in Table 13 (Appendix A.4.8), showing that UPO has the most consistently low resource utilization of these multi-objective methods. Note that showing both time and memory usage for multiple methods and models was challenging given the limited space in the main text (e.g., in Table 5), so we moved this discussion to the appendix.
> - Regarding Weakness 4, we notice that there is no performance drop even with 9 unique rewards. That said, we add discussion about this in Appendix A.4.9 with an experiment involving optimizing the most "noisy" set of rewards possible, which is fully random overall rewards. Though we may expect to see worse generative performance with random rewards, we don't notice any negative impact and explain this through the properties of our RL-based loss function in UPO, wherein a "worst case" random reward leads to noisier SFT-like alignment behaviour.
>
> If you have any additional thoughts on our response, please do let us know. We are happy to discuss and incorporate any additional feedback, and we look forward to hearing from you.

---

> > ### Comment · Reviewer_Ceaz · 2025-03-25
> > **Reply to Authors**
> >
> > Dear Authors,
> >
> > I have reviewed the follow-up response and the revised paper. I appreciate the detailed explanation regarding the questions, which addressed my major concerns. Additionally, I thank you for providing supplementary details to enhance the quality of the papers.

---

### Review · Reviewer_4YqK · 2025-03-25

**Summary Of Contributions:**

The paper's main contribution is UPO (Unified Preference Optimization): a way to have the flexibility of RLHF (in terms of not having to have paired responses) but with less computational requirements and simplicity of MLE objectives like DPO. This enables optimizing over arbitrary preferences, ones that might not be describable through paired preference pairs.

The way the authors enable this is through reformulating optimum policy as one that optimizes preference reward subject to a KL-divergent penalty. The derived UPO objective becomes a unified loss of offline RL and MLE objective. Although still using RL, UPO simplifies the RL objective, resulting in a more computationally efficient method.

**Audience:**

Yes

**Broader Impact Concerns:**

No concerns and potential negative impacts beyond that of traditional LLM alignment and generation potential impact.

**Claims And Evidence:**

Yes

**Requested Changes:**

No requested change

**Strengths And Weaknesses:**

Strengths:
- The motivation for a unified approach that enjoys the benefit of RL and DPO is clear
- Although still using RL, the objective is simplified that it still significantly reduces RLHF complexity
- A clear step towards method that enjoys flexible and powerful alignment, with simplified steps and objective

Weaknesses:
- The use of RL still requires big compute; cost grows exponentially as we tune bigger models. Not a weakness of this work per se, and this work simplifies the traditional RLHF objecttive.
- I notice that very old LMs are being used for experiments here (Llama and Pythia), would be great if we can see results on newer LLMs (Llama 3.2, Qwen, etc)

---

> ### Author Response · Authors · 2025-03-25
> **Response to Reviewer 4YqK**
>
> Thank you for your feedback. Below, we've addressed some of your concerns.
>
> >The use of RL still requires big compute; cost grows exponentially as we tune bigger models. Not a weakness of this work per se, and this work simplifies the traditional RLHF objecttive.
>
> We agree that this is definitely a weakness of RL (on-policy, specifically). However, in contrast, for UPO (Algorithm 1), our RL formulation requires nearly no additional compute relative to the time it takes to complete a forward pass through the transformer to compute logits (required for even SFT). In fact, since it's simply a weighted log probability, as shown in Equation 7, it can be applied on the existing computed logits for the preference method (e.g., DPO/KTO) and requires not even a single additional forward pass through a language model.
>
> We discuss the computational benefits of UPO in Table 5, showing that less than 0.5% of the time spent training is involved with our additional RL component (for 1 day of training, that represents less than 5 minutes, which is negligible). Further, we compare to other competitive RL-based approaches in Table 13 (Appendix A.4.8), where we show that UPO consumes significantly less time/cost than offline PPO and less memory than DRO-V. Across the board, the overall least resource intensive RL technique is UPO and its addition on top of DPO or KTO is negligible in our empirical analyses.
> >I notice that very old LMs are being used for experiments here (Llama and Pythia), would be great if we can see results on newer LLMs (Llama 3.2, Qwen, etc)
>
> We address this in Section 5.1 (end of Page 8) and Appendix A.3.3 (in subsection "Models"). This is a deliberate experimental choice on our part, which is why we decided to justify it in the main text and appendix. Below is a summary of the reasons why we did not perform analyses on any new models (adapted from Sec 5.1/App A.3.3):
>
> - One of our primary concerns with newer models is that they employ many different techniques to “pre-tune” or “pre-align” the models for safety, as demonstrated in these sections. Specifically, LLAMA-3 and other new models like Gemma from Google incorporate significantly greater sets of safety measures, with red teaming, dataset sanitization, and other techniques to ensure safety. Given that safety is one of our alignment tasks, we believe that this is similar to “task leakage” in the training step for these LLMs and makes post-hoc safety alignment more redundant.
> - Many instances of recent work, including Ethyarajh et al. (2024), which is the origin of KTO, or Gao et al. (2024), which is the origin of REBEL, use the original LLAMA and/or PYTHIA models. Note that KTO was introduced at the beginning of 2024, but REBEL was published just 2-3 months ago at NeurIPS 2024. The utility of other work sharing the same models is that it saves a significant amount of computation to retune, retrain, and reevaluate models, which can be extremely expensive (~2,000-4,000 additional GPU hours across models). By leveraging models that the authors train and cross-checking by reproducing some results if needed, we also ensure a fair comparison with the "best case" of these baselines given that we don't need to tune hyperparameters (original authors have already done that to the best of their abilities).
> - We believe that these newer models themselves have (a) not grown significantly larger since 2023 (LLAMA-3 is presented at similar model scales as the original LLAMA) and (b) not changed much given the transformer architecture is largely similar to before. Given that
> emergent behaviour is largely tied to model size, we believe that similar trends exist with older models (though the "baseline" for newer models may be better due to better pre-training procedures and other improvements).
> - There is (amazingly) always a "newer" and "better" model, but given the fast-moving nature of LLMs (though trends regarding model scaling haven't massively shifted in recent years), we sought an "as-fair-as-possible" relative comparison between tried and tested models. Further, we believe that models that are less than 2 years old are still fairly relevant. Though movement is rapid, we still believe it is worth deliberate and careful analysis of models that are still very much used in academic and industry contexts (e.g., the original LLAMA is still ubiquitous, though two major versions have released following its success). In a handful of months, it is plausible that there is another LLM that makes a big "splash" with state-of-the-art metrics in the same way that LLAMA-3, DeepSeek, etc. have, but we still believe such "industry standard" LM approaches that stand the test of time are worth visiting for extended analysis.
>
> Given these reservations we initially had and given the cost (~2K-3K GPU hours) of training with another model, we opted to stick with the major (yet maybe more outdated) LLAMA and PYTHIA models for our work. Please let us know your thoughts about this.

---

> ### Author Response · Authors · 2025-04-05
> **Follow-up to Response**
>
> We hope you had a great week! Thank you again for your constructive feedback on our paper, and we would love to know if you had any additional comments regarding our responses and changes since submission to the computational benchmarking. Please let us know if you have any additional feedback.

---

### Decision · Action_Editor_WEVg · 2025-05-19

**Recommendation:** Accept as is

**Comment:**

This paper studies alignment for large language models. Two of the main families of techniques are RLHF and DPO, where the latter and its variants are cheaper but harder to control. The authors introduce a simple approach that gets the best properties of both (or, rather, avoids the bad properties of either, e.g., the cost of on-policy RL for RLHF). They introduce a new variation that combines MLE and an auxiliary/reward objective. They have a nice theoretical derivation for particular instantiations of the preference model and achieve strong experimental results.

The reviewers are positive. The most common questions were about clarity and asking for a little bit more experimental evidence. The authors did a solid job; the paper is quite readable. Ultimately, everyone (including me) agreed that this is a clever approach that balances all the major challenging factors that are needed for alignment.

**Audience:**

Yes, alignment in large language models is an important area for the TMLR audience.

**Claims And Evidence:**

Yes, the paper has analysis and experimental evidence for the claims made.

---

> ### Author Response · Authors · 2025-05-24
> **Thank You & Submission**
>
> We would like to thank you and all the reviewers for the feedback and hard work, which has really improved our work. We're delighted to have our work accepted at TMLR, and we believe it will benefit the research community. From our side, we have uploaded the camera ready submission of our work. Thank you all again!